# Effect of Dietary Riboflavin Levels on Reproductive Performance of Pigeon Breeders, and Growth Performance and Carcass Traits of Offspring Squabs

**DOI:** 10.3390/ani14162414

**Published:** 2024-08-20

**Authors:** Bo Zhang, Yusheng Gao, Yuxin Shao, Li Shen, Wenli Liu, Haoxuan Li, Yipu Li, Jing Li, Tenghe Ma, Zheng Wang

**Affiliations:** 1Institute of Animal Husbandry and Veterinary Medicine, Beijing Academy of Agriculture and Forestry Sciences, Beijing 100097, China; zhangb950414@163.com (B.Z.); 18832004998@163.com (Y.G.); syu123@sina.com (Y.S.); 17320897356@163.com (L.S.); 15569845062@163.com (W.L.); crystalnihility@163.com (H.L.); 18730978250@163.com (Y.L.); xsu1980@163.com (J.L.); 2School of Life Science and Food Engineering, Hebei University of Science and Technology, Shijiazhuang 050091, China; matenghe@126.com

**Keywords:** pigeon breeder, riboflavin, reproductive performance, growth performance, carcass trait

## Abstract

**Simple Summary:**

Pigeons are one of the oldest domesticated birds in the world, and humans have been raising pigeons for thousands of years. Pigeon meat is edible and rich in protein, polyunsaturated fatty acids, minerals, and vitamins. As a late-maturing bird, pigeons need to be fed by their parents to obtain nutrients. Therefore, the dietary nutritional level of pigeon breeders is vital for production and reproduction. However, there is currently no explicit nutritional requirement for pigeon riboflavin internationally.

**Abstract:**

This study aimed to investigate the effects of dietary riboflavin levels on the reproductive performance of pigeon breeders and the growth performance and carcass traits of offspring squabs to estimate the riboflavin requirement of pigeon breeders. The natural riboflavin content in the basic diet of corn–peas–soybean–wheat–sorghum–corn gluten is 1.20 mg/kg. Different doses of riboflavin (0, 2.5, 5, 10, and 15 mg/kg) were supplemented with the basal diet to produce five dietary treatments with total riboflavin levels of 1.20, 3.70, 6.20, 10.20, and 16.20 mg/kg. A total of 120 pairs of White King pigeons, aged 60 wks, were randomly allocated into five treatment groups, each consisting of 24 pairs. Each pair was individually raised for 8 wks. After the experiment, an assessment was conducted to evaluate the reproductive performance of the pigeon breeders, as well as the growth and carcass traits of offspring squabs at 28 days of age. The results showed that the dietary riboflavin levels had no significant effect on body weight, feed intake, egg weight, egg production, and egg fertility (*p* > 0.05). However, pigeons fed a diet without riboflavin had the lowest egg hatchability, egg yolk color, carcass trait, and riboflavin status, while exhibiting higher liver weight and liver index (*p* < 0.05). Moreover, the indices above showed increased or decreased linearly as the level of riboflavin was increased in the diet. Based on the broken-line regression model, pigeon breeders were determined to require a dietary riboflavin content of 11.4, 13.6, 13.4, 6.60, 4.28, 4.47, 4.67, 6.69, and 6.82 mg/kg to optimize hatchability, eviscerated weight, half-eviscerated weight, breast muscle weight, breast muscle percentage, liver weight, liver index, egg yolk riboflavin, and squab plasma riboflavin, respectively. In conclusion, the optimal supplemental dosage of riboflavin in the diets of pigeon breeders is 13.6 mg/kg.

## 1. Introduction

Pigeons are one of the oldest domesticated birds in the world, and humans have been raising pigeons for thousands of years. Pigeon meat is edible and rich in protein, polyunsaturated fatty acids, minerals, and vitamins [1]. China is the largest producer and consumer of pigeon meat in the world. In 2022, the country had a total of 43.86 million pairs of breeding pigeons and produced 440 million squabs. Pigeons have become the fourth most important poultry category in China, following chickens, ducks, and geese. Unlike other poultry, pigeons are considered late-maturing birds that require “nursing” from breeding pigeons to provide pigeon milk to meet their nutritional needs for growth and development. Inadequate nutrition supply in their feed can affect the breeding performance of parent birds, reduce offspring survival, and lead to poor development of squabs.

Riboflavin, a water-soluble vitamin B, acts as a precursor for flavin mononucleotide and flavin adenine dinucleotide. The liver serves as the primary reservoir for riboflavin and plays a significant role in its metabolism. Given that humans and monogastric animals are unable to synthesize sufficient riboflavin to satisfy tissue demands, dietary intake becomes essential to meet their physiological needs [2,3]. Insufficient intake of riboflavin in the diet will produce riboflavin deficiency. Riboflavin deficiency has negative effects on humans and animals, such as growth and development obstruction [4,5], poor reproductive performance [5,6,7], and tissue and organ damage [4,8]. For laying poultry, riboflavin deficiency in laying hens reduces body weight, egg weight, egg production, and egg hatchability, and increases chicken embryo mortality during incubation [6,7]. Chick embryos with hereditary riboflavin deficiency exhibit abnormal development and experience sudden death during the mid-incubation period [9,10,11]. Similarly, ducks exhibit comparable outcomes when subjected to a riboflavin-deficient diet, as evidenced by a gradual decline in egg hatchability approaching zero over an extended feeding period, alongside diminished concentrations of riboflavin in both plasma and eggs [5,12]. Concerning pigeons, a universally accepted set of criteria is currently absent, and breeding practices exhibit substantial disparities across various geographical areas. Furthermore, a dearth of research exists concerning the riboflavin needs of pigeons. Consequently, the objective of this study is to examine the effect of dietary riboflavin levels on the reproductive performance of pigeon breeders and the productivity performance and carcass traits of offspring squabs. Ultimately, this study assesses the riboflavin requirement of pigeon breeders.

## 2. Materials and Methods

### 2.1. Experimental Design

Five riboflavin gradients (1.20, 3.70, 6.20, 11.2, and 16.2 mg/kg) of the dose–response experiment were conducted to study the effects of riboflavin on the reproductive performance of pigeon breeders and the growth performance and carcass traits of offspring squabs to evaluate the nutritional requirement of riboflavin in pigeon breeders. The basal full pellet diet with corn–peas–soybean–wheat–sorghum–corn gluten diet (Table 1) containing 1.20 mg/kg of naturally occurring riboflavin was supplemented with 0, 2.50, 5, 10, and 15 mg/kg of riboflavin in the diet, resulting in total riboflavin concentrations of 1.20, 3.70, 6.20, 11.2, and 16.2 mg/kg, respectively. A total of 120 pairs of White King pigeons aged 60 wks were randomly divided into five treatment groups with twenty-four replicates (pairs), receiving the above five experimental diets for 8 wks. During the experiment, each pair of pigeon breeders was individually housed in separate feed cages (50 cm length × 50 cm deep × 60 cm height). Meanwhile, each breeding pair mated freely and received ad libitum access to water and diet. The pigeons received eighteen hours of light per day from 04:00 to 22:00 h. The eggs in each litter were incubated by the parents.

### 2.2. Experimental Diets

The riboflavin concentration in the basal diet was determined by reversed-phase high-performance liquid chromatography (HPLC). The determined method was referred to in a previous study [13], and the feed samples were prepared according to the method described previously [14,15]. Except for riboflavin content, the nutritional components of the basal diet meet the nutritional requirements of pigeon breeders (China Animal Husbandry and Veterinary Association, 2020). All diets were cold-pelleted at room temperature. Crystal riboflavin (99% purity) was purchased from Sigma Aldrich (St. Louis, MO, USA).

### 2.3. Productivity and Reproductive Performance

All pigeon breeders were weighed at the beginning and end of the experiment. During the experiment, the eggs were recorded daily to calculate egg production on a treated replicate basis. On the fifth day of incubation, fertile eggs were calculated by candling the eggs, and egg fertility was calculated on a treated replicate basis. On the 18th day of incubation, the brood was recorded to obtain the hatchability of fertile eggs and calculated egg hatchability on a treated replicate basis.

### 2.4. Egg Quality

At the end of the experiment, 12 pigeon eggs were randomly selected from each group (*n* = 12) for the assessment of egg quality. The eggshell color was evaluated using an eggshell color tester (QCR, TSS, York, UK) and reflectance was utilized to represent the color. The strength of the eggshells was measured with an egg force reader (EFR-01, ORKA, Tel Aviv, Israel). Following the weighing of the eggs, their horizontal and vertical diameters, along with eggshell thickness, were determined using a digital caliper. When measuring eggshell thickness, the eggshell membrane was initially removed, and measurements were taken at the blunt end, pointed end, and equatorial end of the egg, with the average value calculated. Egg weight, yolk color, albumen height, and Haugh unit were analyzed using an automated egg quality analyzer (EMT7300, Robotmation, Tokyo, Japan). The yolks were separated, weighed, and stored at −20 °C for subsequent yolk riboflavin concentration measurement.

### 2.5. Growth Performance, Carcass Trait, and Organ Index

During the days 1, 7, 14, and 28 of offspring squabs, 12 birds from each replicate were weighted (*n* = 12). Subsequently, at 28 days of age, 12 randomly selected squabs from each treatment group were weighed and then slaughtered by neck cutting and manual evisceration. The boneless and skinless breast muscles (pectoralis major and pectoralis minor), leg muscles (thigh muscles and drumstick muscles), and abdominal fat were manually removed. The following indicators were weighted: eviscerated carcass, half-eviscerated carcass, breast meat, leg meat, liver, spleen, and bursa. To compensate for differences in pigeon body size, the values were expressed as a percentage of body weight (eviscerated, half-eviscerated, liver, spleen, and bursa) or eviscerated carcass weight (breast muscle and leg muscle) according to Chinese standards [13].

### 2.6. Riboflavin Concentration

The riboflavin concentration in the basal diet, plasma, egg yolk, and liver were determined by reversed-phase high-performance liquid chromatography (HPLC). The determined method was referred to in a previous study [14]. Before being detected by HPLC, feed and plasma samples were prepared according to the method described previously [15,16], whereas the egg yolk and liver samples were prepared according to the method for animal tissue described in a previous study [17]. Except for riboflavin content, the nutritional components of the basal diet meet the nutritional requirements of pigeon breeders (China Animal Husbandry and Veterinary Association, 2020). All diets were cold-pelleted at room temperature. Crystal riboflavin, flavin mononucleotide (FMN), and flavin adenine dinucleotide (FAD) (99% purity) were purchased from Sigma Aldrich (St. Louis, MO, USA).

### 2.7. Statistical Analysis

Data were tested for normal distribution (Gaussian) using the Shapiro–Wilk normality test and homogeneity of variances by Levene’s test. Data with a normal distribution (Shapiro–Wilk test > 0.05) were examined by a one-way ANOVA followed by Duncan’s multiple comparisons (SAS 9.4, 2011, SAS Institute Inc.). Data that did not follow a normal distribution (Shapiro–Wilk test ≤ 0.05) were analyzed using the Kruskal–Wallis test (SAS 9.4, *p* < 0.05 indicated statistical significance), and if the result was statistically significant, the Dwass–Steel–Critchlow–Fligner (DSCF) test was used for pairwise comparison. All results were examined for quadratic and linear effects by orthogonal polynomial contrast, with significance declared at *p* ≤ 0.05. The reported *p*-values were Benjamini–Hochberg-corrected. *p*  <  0.05 was regarded as statistically significant. The results are expressed as the mean value and standard error of the mean (SEM).

A broken-line regression analysis [18] was used to estimate the riboflavin requirement of pigeon breeders. The model formulation was provided as follows:*y* = *l* + *u* (*r* − *x*)
where *y* = egg hatchability, eviscerated weight, half-eviscerated weight, breast muscle weight, breast muscle percentage, liver weight, liver index, egg yolk riboflavin, or squab plasma riboflavin; *x* = dietary total riboflavin level (mg/kg); *r* = riboflavin requirement of pigeon breeders; *l* = the response at *x* = *r*; and *u* = the slope of the curve. In the model, *y* = *l* when *x* > *r*.

## 3. Results

### 3.1. Reproductive Performance

In this study, dietary riboflavin levels did not affect the body weight of pigeon breeders, feed intake, laying interval, egg production, and egg fertility (*p* > 0.05, Table 2). However, at 8 wks into the experiment, the pigeons fed the diet without the riboflavin supplement had the lowest egg hatchability (59.87%), and a linear or quadratic increase was observed as the level of riboflavin was increased in the diet (*p* < 0.05, Table 2). Moreover, the egg hatchability increased as dietary riboflavin was increased, and reached a plateau when dietary riboflavin was above 11.4 mg/kg (y = 77.6 − 1.92 × (11.4 − x), *p*-value = 0.036, *R*^2^ = 0.964).

### 3.2. Egg Quality

As shown in Table 2, pigeon egg weight, egg shape index, eggshell color, eggshell strength, albumen height, Haugh unit, egg yolk weight, eggshell weight, and eggshell thickness were not affected by the level of dietary riboflavin concentration (*p* > 0.05, Table 3). However, the egg yolk color increased linearly as the level of riboflavin was increased in the diet (*p* < 0.05, Table 3 and Figure 1).

### 3.3. Growth Performance and Carcass Trait

It was observed that the dietary riboflavin levels did not have a significant effect on the body weight at 7, 14, and 28 days, as well as on carcass weight, eviscerated percentage, half-eviscerated weight, leg muscle weight, and leg muscle percentage of squabs at 28 days of age (*P* or *Z* > 0.05, as shown in Table 4 and Table 5). The effect of dietary riboflavin levels on the carcass traits of squabs is shown in Table 5. It is worth noting that, without the riboflavin supplement, the groups had lower birth weight, as well as lower eviscerated weight and breast muscle weight; moreover, the breast muscle percentages of squabs at 28 days were all lower than those in the riboflavin supplement groups (*P* or *Z* < 0.05, Table 4 and Table 5), and half-eviscerated weight showed a decreasing tendency (0.05 < *p* < 0.10, Table 5). Moreover, the above indicators increased linearly as the level of riboflavin was increased in the diet, and reached a plateau when dietary riboflavin was above 13.6 (y = 368 − 4.68 × (13.6 − x), *p*-value = 0.088, *R*^2^ = 0.912), 13.4 (y = 397 − 4.91 × (13.4 − x), *p*-value = 0.034, *R*^2^ = 0.966), 6.60 (y = 397 − 4.91 × (13.4 − x), *p*-value = 0.098, *R*^2^ = 0.902), and 4.28 mg/kg (y = 23.6 − 1.27 × (4.28 − x), *p*-value = 0.013, *R*^2^ = 0.987), respectively.

### 3.4. Organ Weight and Index

The effect of dietary riboflavin levels on organ weight and organ index is shown in Table 6. The spleen weight, spleen index, bursa weight, and bursa index of offspring squabs were all not affected by the content of dietary riboflavin. However, the liver weight and liver index in the group without the riboflavin supplement were significantly higher than those in the riboflavin supplement groups. Meanwhile, the above indicators decreased linearly and quadratically as the level of riboflavin was increased in the diet, and reached a plateau when dietary riboflavin was 4.47 (y = 12.9 + 1.13 × (4.47 − x), *p*-value = 0.012, *R*^2^ = 0.988) and 4.67 mg/kg (y = 2.55 + 0.258 × (4.67 − x), *p*-value = 0.025, *R*^2^ = 0.975), respectively.

### 3.5. Riboflavin Status

As shown in Table 7, the pigeon fed the basal diet without riboflavin showed the lowest riboflavin concentration in female pigeon breeder plasma, male pigeon breeder plasma, egg yolk, squab plasma, and squab liver. Meanwhile, squab liver FMN and FAD concentration were also the lowest in the group without riboflavin. All these parameters increased linearly and quadratically as the level of riboflavin was increased in the diet (Table 7). Specifically, egg yolk and squab plasma riboflavin concentration reached a plateau when the dietary riboflavin was above 6.69 (y = 14.6 − 1.40 × (6.69 − x), *p*-value < 0.001, *R*^2^ = 0.999) and 6.82 mg/kg (y = 2.56 − 0.318 × (6.82 − x), *p*-value < 0.001, *R*^2^ = 0.999), respectively.

## 4. Discussion

Riboflavin is essential for poultry production and reproduction. Recommended levels for chickens are 3.6 mg/kg [19] and 8 mg/kg [20]. Ducks require 4 mg/kg [19], 10 mg/kg for meat ducks [21], and 15 mg/kg for duck breeders [21]. As a late maturing bird, pigeons need to be fed by their parents to obtain nutrients. Therefore, the dietary nutritional level of pigeon breeders is vital for production and reproduction. However, there is currently no explicit nutritional requirement for pigeon riboflavin internationally. This study aimed to investigate the effect of dietary riboflavin levels on the reproductive performance of pigeon breeders and the growth and carcass traits of offspring squabs to evaluate the riboflavin requirement of pigeon breeders.

The NRC in 1971 found that riboflavin deficiency in chickens can cause symptoms like diarrhea, growth retardation, and roll paw paralysis. Research has shown that without riboflavin in their diet, 10% of chicks develop roll paw paralysis and a 25% mortality rate [22]. Studies on Pekin ducks have also shown that riboflavin deficiency can lead to reduced retarded growth, poor carcass traits, and liver injury [8,13,23]. Adequate dietary riboflavin is essential for the reproductive performance of laying poultry. Egg hatchability significantly decreased to 5% in laying hens fed a riboflavin-deficient diet for 3 wks compared to over 80% in the riboflavin-sufficient group [6,7]. Furthermore, a strain of Leghorn chicken with a genetic deficiency in riboflavin deposition in their eggs produces riboflavin-deficient eggs, resulting in the death of the embryos during the mid-incubation period [9,24,25]. Consistent with previous research, the results of this study indicate that the group without the riboflavin supplement had the lowest egg hatchability, and this value increased linearly or quadratically as dietary riboflavin increased.

The broken-line regression analysis has been widely used to assess riboflavin requirements in chickens [26] and ducks [12,13,27]. Therefore, in the results of this study, the regression analysis method was also applied to evaluate the riboflavin requirement of pigeon breeders. According to the regression analysis, the riboflavin requirement of pigeon breeders for egg hatchability was 11.4 mg/kg in diet, which is close to the riboflavin requirement value of duck breeders (Ministry of Agriculture of People’s Republic China, 2012).

Maternal nutrition is the key to maintaining the normal growth and development of offspring [28,29]. Studies have shown that riboflavin deficiency in pregnant rats decreased plasma riboflavin concentration and birth weight of newborn rats [29,30]. Poultry transfer nutrients to the eggs through the maternal plasma, and during the incubation process, avian embryos continuously absorb nutrients from the egg (primarily egg yolk) [31,32,33]. Tissue riboflavin is considered to be a useful biomarker for riboflavin status in animals [34,35,36,37]. In previous studies on poultry, riboflavin deficiency has been found to result in reduced plasma and egg riboflavin concentration, which impaired embryonic development and even caused mortality in offspring [4,5,12]. Offspring ducklings hatched from riboflavin-deficient groups exhibit lower growth performance after being supplemented with adequate riboflavin compared to offspring with sufficient riboflavin levels [38]. Unlike chickens, ducks, and geese, pigeons not only rely on the nutrients transmitted from the maternal plasma to the egg during the hatching period but also need to obtain nutrients through parental feeding. Therefore, the level of riboflavin in the pigeon diet is vital. In the present study, the offspring of the group without riboflavin addition had significantly lower riboflavin status and poor carcass traits than those with riboflavin addition, which increased linearly or quadratically and reached a plateau. The regression analysis shows that the riboflavin requirements of pigeons for eviscerated weight, half-eviscerated weight, breast muscle weight, breast muscle percentage, egg yolk riboflavin concentration, and squab plasma riboflavin were 13.6, 13.4, 6.60, 4.28, 6.69, and 6.82 mg/kg, respectively.

Riboflavin is stored in the liver and plays a crucial role in liver function. Studies have shown that riboflavin deficiency induces enlargement of livers and increased hepatic total fat, triglycerides, and cholesterol levels [39,40,41,42,43]. In chicken embryos, riboflavin deficiency can cause liver discoloration and accumulation of fat and lipid metabolism products [4]. In the present study, riboflavin deficiency resulted in hepatic swelling, a condition that was ameliorated by riboflavin supplements. Regression analysis showed that riboflavin requirements for liver weight and liver index were 4.47 and 4.67 mg/kg, respectively.

## 5. Conclusions

In conclusion, pigeon diet riboflavin deficiency reduced egg hatchability, egg quality, riboflavin status, and squab carcass trait. However, these adverse effects can be prevented by providing sufficient riboflavin to the pigeons. According to the broken-line regression analysis, the riboflavin requirements of pigeons for hatchability, eviscerated carcass weight, half-eviscerated carcass weight, breast muscle weight, breast muscle ratio, liver weight, liver index, egg yolk riboflavin, and squab plasma riboflavin were 11.4, 13.6, 13.4, 6.60, 4.28, 4.47, 4.67, 6.69, and 6.82 mg/kg, respectively. In the present study, the recommended riboflavin requirement in pigeon breeder diets is 13.6 mg/kg.

## Figures and Tables

**Figure 1 animals-14-02414-f001:**
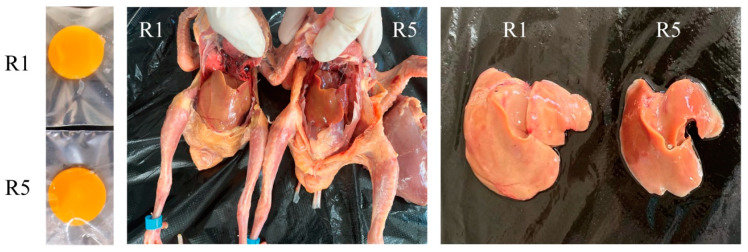
Effects of dietary riboflavin levels on the egg yolk and liver of squabs. R1—no additional riboflavin was added to the basal diet; R5—a further 15 mg/kg of riboflavin was added to the basal diet.

**Table 1 animals-14-02414-t001:** Composition of riboflavin-deficient basal diet (g/kg as-fed basis).

Ingredient	Content
Corn	400.0
Soybean meal (44% CP)	68.0
Pea	180.0
Wheat	128.0
Corn gluten meal (50% CP)	50
Sorghum	104.0
Soybean oil	7.0
Dicalcium phosphate	22.3
Limestone	25
Sodium chloride	4.0
Vitamin premix ^1^	0.3
Choline chloride	0.4
Mineral premix ^2^	1.0
DL-Methionine	6.0
Lysine hydrochloride	4.0
Calculated composition	
Metabolizable energy, Mcal/kg	2.99
Crude protein ^3^	159.1
Calcium	15.2
Non-phytate phosphorus	5.4
Lysine	6.4
Methionine	2.5
Methionine + cysteine	5.0
Threonine	5.3
Tryptophan	1.5
Arginine	9.5
Riboflavin ^4^, mg/kg	1.20

^1^ Complex vitamins were provided per kilogram of feed: VA 13,500 IU, VD_3_ 3600 IU, VE 36 IU, VK_3_ 5 mg, VB_1_ 5.0 mg, D-pantothenic acid 22 mg, nicotinamide 41.94 mg, folic acid 3 mg, biotin 0.25 mg, and VB_12_ 0.039 mg. ^2^ Trace components were provided per kilogram of feed: iron 150 mg, zinc 88 mg, copper 4 mg, manganese 70 mg, iodine 0.35 mg, and selenium 0.25 mg. ^3^ The nutrient compositions are calculated values. ^4^ The values were analyzed by high-performance liquid chromatography.

**Table 2 animals-14-02414-t002:** Effects of dietary riboflavin levels on reproductive performance of pigeon breeders.

Dietary Riboflavin (mg/kg)	1.20	3.70	6.20	11.2	16.2	*SEM*	*F*-Value	*H*-Value	*p*-Value	*p*-Value Adjustment	*p*-Value Riboflavin Linear	*p*-Value Riboflavin Quadratic
Initial body weight (g/pair) ^1^	1182	1177	1189	1188	1202	6.86	0.363	--	0.956	1.00	0.884	0.717
Final body weight (g/pair) ^1^	1237	1217	1252	1203	1182	13.7	0.901	--	0.934	1.00	0.902	0.851
Total feed intake of nursing (g/pair·28d) ^1^	3908	3910	3908	3909	3907	2.30	0.081	--	0.667	0.840	0.785	0.693
Laying interval (d) ^1^	39.0	40.7	40.1	43.2	40.8	0.703	1.14	--	0.305	0.373	0.476	0.356
Egg production (%) ^1^	90.2	93.5	92.3	93.2	90.9	0.860	0.329	--	0.883	0.946	0.562	0.679
Egg fertility (%) ^1^	92.3	98.8	94.7	94.4	93.9	0.553	0.733	--	0.945	1.00	0.895	0.956
Egg hatchability (%) ^2^	59.9b	60.1 b	67.9 ab	77.7 a	77.6 a	0.044	--	17.15	0.002	0.018	0.019	0.022

Results are means with *n* = 24 per group. a,b Means with different superscripts within the same column differ significantly (*p* < 0.05). ^1^ Data were examined by a one-way ANOVA followed by Duncan’s multiple comparisons. ^2^ Data were analyzed using the Kruskal–Wallis test, and the Dwass-Steel–Critchlow–Fligner (DSCF) test was used for pairwise comparison.

**Table 3 animals-14-02414-t003:** Effects of dietary riboflavin levels on pigeon egg quality ^1^.

Dietary Riboflavin (mg/kg)	1.20	3.70	6.20	11.2	16.2	*SEM*	*F*-Value	*H*-Value	*p*-Value	*p*-Value Adjustment	*p*-Value Riboflavin Linear	*p*-Value Riboflavin Quadratic
Egg weight (g)	24.5	23.0	24.5	24.6	23.8	0.241	1.56	--	0.193	0.354	0.984	0.602
Egg shape index (vertical diameter, cm)	43.5	43.0	43.6	44.0	43.3	0.172	0.726	--	0.577	0.907	0.709	0.402
Egg shape index (horizontal diameter, cm)	31.9	31.4	32.0	31.9	32.2	0.173	0.599	--	0.664	0.913	0.306	0.780
Eggshell color (%)	66.2	66.2	65.6	64.7	66.5	0.825	0.377	--	0.484	1.00	0.862	0.298
Eggshell strength (kg)	1.19	1.24	1.11	1.18	1.13	0.021	1.15	--	0.342	0.577	0.272	0.860
Egg albumen height	3.22	2.84	3.18	2.66	2.62	0.104	1.54	--	0.203	0.319	0.051	0.920
Egg yolk color	7.28 b	7.68 b	7.94 ab	8.71 a	8.78 a	0.152	4.49	--	0.003	0.017	<0.001	0.356
Haugh units	67.1	57.3	57.9	56.7	60.7	2.12	0.938	--	0.447	1.00	0.512	0.148
Egg yolk weight (g)	4.04	3.90	4.37	3.81	3.69	0.121	0.866	--	0.488	1.00	0.262	0.455
Eggshell weight (g)	4.83	5.39	4.79	5.56	5.13	0.136	1.19	--	0.323	0.395	0.416	0.440
Eggshell thickness (mm)	0.250	0.262	0.262	0.243	0.247	0.003	1.58	--	0.187	0.411	0.175	0.463

Results are means with *n* = 12 per group. a,b Means with different superscripts within the same column differ significantly (*p* < 0.05). ^1^ Data were examined by a one-way ANOVA followed by Duncan’s multiple comparisons.

**Table 4 animals-14-02414-t004:** Effects of dietary riboflavin levels on body weight of squabs at different ages.

Dietary Riboflavin (mg/kg)	1.20	3.70	6.20	11.2	16.2	*SEM*	*F*-Value	*H*-Value	*p*-Value	*p*-Value Adjustment	*p*-Value *Riboflavin Linear*	*p*-Value *Riboflavin Quadratic*
1 d BW (g) ^2^	19.2 b	19.3 b	20.8 ab	21.7 a	21.9 a	0.316	--	17.15	0.003	0.018	0.002	0.371
7 d BW (g) ^2^	140	139	148	144	171	0.673	--	2.602	0.521	0.627	0.203	0.523
14 d BW (g) ^1^	361	356	362	363	381	3.90	1.24	--	0.303	0.682	0.050	0.496
28 d BW (g) ^1^	459	440	519	512	522	14.8	1.62	--	0.169	0.466	0.149	0.502

BW: body weight. The results are means with *n* = 12 per group. a,b Means with different superscripts within the same column differ significantly (*p* < 0.05). ^1^ Data were examined by a one-way ANOVA followed by Duncan’s multiple comparisons. ^2^ Data were analyzed using the Dwass–Steel–Critchlow–Fligner (DSCF) test for pairwise comparison.

**Table 5 animals-14-02414-t005:** Effects of dietary riboflavin levels on carcass traits of offspring squabs.

Dietary Riboflavin (mg/kg)	1.20	3.70	6.20	11.2	16.2	*SEM*	*F*-Value	*H*-Value	*p*-Value	*p*-Value Adjustment	*p*-Value Riboflavin Linear	*p*-Value Riboflavin Quadratic
Carcass weight (g) ^1^	412	404	418	417	426	6.34	0.261	--	0.827	0.902	0.440	0.854
Eviscerated weight (g) ^2^	319 b	310 b	336 ab	359 a	368 a	5.78	--	8.99	0.005	0.024	0.106	0.439
Eviscerated percentage (%) ^2^	66.3	66.4	67.1	67.3	69.3	0.693	--	3.83	0.227	0.421	0.238	0.716
Half-eviscerated weight (g) ^1^	342	341	363	387	397	6.05	3.10	--	0.018	0.066	0.009	0.680
Half-eviscerated percentage (%) ^2^	69.9	71.8	72.5	74.8	74.6	0.637	--	5.76	0.115	0.218	0.211	0.835
Breast muscle weight (g) ^2^	59.5 b	72.7 a	74.5 a	77.1 a	78.2 a	1.92	--	10.4	0.002	0.015	0.029	0.202
Breast muscle percentage (%) ^2^	19.7 b	22.9 a	23.4 a	23.5 a	24.0 a	0.357	--	14.8	<0.001	0.005	0.044	0.107
Leg muscle weight (g) ^1^	21.5	22.0	22.6	23.4	24.8	0.432	1.26	--	0.287	0.395	0.054	0.585
Leg muscle percentage (%) ^1^	6.82	7.04	6.76	6.78	6.74	0.0980	0.290	--	0.884	0.912	0.517	0.941

Results are means with *n* = 12 per group. a,b Means with different superscripts within the same column differ significantly (*p* < 0.05). ^1^ Data were examined by a one-way ANOVA followed by Duncan’s multiple comparisons. ^2^ Data were analyzed using the Kruskal–Wallis test, and the Dwass–Steel–Critchlow–Fligner (DSCF) test was used for pairwise comparison.

**Table 6 animals-14-02414-t006:** Effects of dietary riboflavin levels on organ index of offspring squabs ^1^.

Dietary Riboflavin (mg/kg)	1.20	3.70	6.20	11.2	16.2	*SEM*	*F*-Value	*H*-Value	*p*-Value	*p*-Value Adjustment	*p*-Value *Riboflavin Linear*	*p*-Value *Riboflavin Quadratic*
Liver weight (g)	16.6 a	13.7 b	13.1 b	12.7 b	12.7 b	0.345	--	16.5	<0.001	0.002	0.005	0.021
Liver index (%)	3.44 a	2.80 b	2.62 b	2.58 b	2.45 b	0.066	--	18.2	<0.001	0.001	<0.001	0.008
Spleen weight (g)	0.406	0.442	0.434	0.454	0.448	0.023	--	0.345	0.826	0.987	0.654	0.737
Spleen index (%)	0.0854	0.0885	0.0861	0.0912	0.0838	0.004	--	0.384	0.817	0.984	0.963	0.696
Bursa weight (g)	0.569	0.713	0.708	0.718	0.841	0.037	--	5.55	0.158	0.236	0.100	0.862
Bursa percentage (%)	0.115	0.139	0.141	0.142	0.158	0.006	--	4.05	0.333	0.400	0.222	0.646

Results are means with *n* = 12 per group. a,b Means with different superscripts within the same column differ significantly (*p* < 0.05). ^1^ Data were analyzed using the Kruskal–Wallis test, and the Dwass–Steel–Critchlow–Fligner (DSCF) test was used for pairwise comparison.

**Table 7 animals-14-02414-t007:** Effects of dietary riboflavin levels on riboflavin content in parent pigeons, eggs, and offspring squabs ^1^.

Dietary Riboflavin (mg/kg)	1.20	3.70	6.20	11.2	16.2	*SEM*	*F*-Value	*H*-Value	*p*-Value	*p*-Value Adjustment	*p*-Value *Riboflavin Linear*	*p*-Value *Riboflavin Quadratic*
Parental plasma (♀, μmol/L)	0.947 d	2.47 c	3.50 b	4.09 a	4.25 a	0.199	--	36.2	<0.001	<0.001	<0.001	<0.001
Parental plasma (♂, μmol/L)	0.546 d	0.918 c	1.35 b	1.63 a	1.66 a	0.070	--	35.1	<0.001	0.004	<0.001	<0.001
Egg yolk (μg/g)	6.93 c	10.5 b	13.9 a	14.5 a	14.7 a	0.486	--	32.5	<0.001	<0.001	<0.001	<0.001
Squab plasma (μmol/L)	0.765 c	1.56 b	2.36 a	2.55 a	2.56 a	0.115	--	31.9	<0.001	<0.001	<0.001	<0.001
Squab liver riboflavin (μg/g)	2.59 c	3.68 b	4.21 a	4.33 a	4.39 a	0.111	--	31.6	<0.001	<0.001	<0.001	<0.001
Squab liver FAD (μg/g)	0.844 d	0.949 c	1.29 b	1.45 a	1.52 a	0.045	--	34.2	<0.001	<0.001	<0.001	<0.001
Squab liver FMN (μg/g)	2.61 d	3.92 c	4.35 b	4.58 a	4.66 a	0.124	--	34.3	<0.001	<0.001	<0.001	<0.001

♀: Female parent pigeon; ♂: male parent pigeon. The results are means with *n* = 8 per group. a–d Means with different superscripts within the same column differ significantly (*p* < 0.05). ^1^ Data were analyzed using the Kruskal–Wallis test, and the Dwass–Steel–Critchlow–Fligner (DSCF) test was used for pairwise comparison.

## Data Availability

Data are contained within the article/Appendix A.

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
