# Peer review of "Effect of Dietary Riboflavin Levels on Reproductive Performance of Pigeon Breeders, and Growth Performance and Carcass Traits of Offspring Squabs"

_animals, 2024, doi:10.3390/ani14162414_

Round 1
Reviewer 1 Report
Comments and Suggestions for Authors
The study reported in this submission appears to have been conducted with due care and the methodology is well described. A good scientific contribution to pigeon nutrition. The manuscript is easy to read and to understand. But still there are excessive number of minor syntax errors. Careful proofreading is strongly recommended.
L3: … breeders and, growth performance…
L10: there are 10 authors. It is therefore unclear what meant by equal contribution?
L14: .. as a late adult birds ….?. revise.
L81: what was the source of riboflavin?. Manufacturer, location, purity etc.
L83: how were the breeding pairs selected?. Randomly? Expand
L86: ..each breeding pair mated freely?. Not clear. Revise
L88: ..each litter? Revise.
A MAJOR ISSUE: L150: Statistical analyses – the data were analyzed using one-way ANOVA, and means were compared using the Duncan multiple range test. This is NOT the appropriate analysis. The appropriate statistical method for this design is orthogonal polynomials to examine the linear and quadratic effects of graded levels of riboflavin. As such, differences between individual treatments are NOT important. The authors are using superscripts to separate significance between riboflavin level means, which is not valid. In this statistical model, trends are relevant than treatment differences. Using linear or quadratic effects would have enabled a much focussed presentation. The data should have been be presented accordingly in the Tables and, the 'ABSTRACT", 'RESULTS" and 'DISCUSSION' sections should be re-written. This is a FATAL flaw and must be addressed.
ALL TABLES: individual treatment means +/- SD are not needed. Show the ‘Pooled SEM’ along with L and Q effects. ALSO The numbers and decimals in tables should be follow the rule of: xxxx, xxx, xx.x, x.xx, 0.xxx and 0.0xxx.
TABLE 4 and 5: Absolute weights (g) are irrelevant –these are related to BW. Delete. Give only the % BW data
FIGURE 1. Present as a Table – why the RB units are different?
ALL text need to be revised and discussed as per L and Q responses- NOT by comparing treatment means
Comments on the Quality of English Language
Moderate revision needed
Author Response
The study reported in this submission appears to have been conducted with due care and the methodology is well described. A good scientific contribution to pigeon nutrition. The manuscript is easy to read and to understand. But still there are excessive number of minor syntax errors. Careful proofreading is strongly recommended.
Response: Thank you very much for your suggestion, it will be crucial for the completion of our manuscript. We have made modifications to the question you raised in the manuscript, and highlighted them in red font for your review. Details as below.
L3: … breeders and, growth performance…
Response: Thanks for your suggestion. We modify this part as “Effects of dietary riboflavin level on reproductive performance of pigeon breeders, and growth performance and carcass trait of offspring squabs”. Please see the details in the manuscript in line 1~3.
L10: there are 10 authors. It is therefore unclear what meant by equal contribution?
Response: Regarding the authors, we would like to state that the highlighted authors B. Zhang and Y.S. Gao contributed equally as the co-first authors of the article.
L14: .. as a late adult birds ….?. revise.
Response: Thanks for your suggestion. We modify this part as “late maturing bird” for a more accurate description. Please see the details in the manuscript in line 14.
L81: what was the source of riboflavin?. Manufacturer, location, purity etc.
Response: Thanks for your suggestion. Crystal riboflavin (99% purity) is purchased from Sigma Aldrich (St. Louis, MO, USA). For a more accurate description, this aspect is detailed in the Materials and Methods - Experimental Diets section. Please refer to the manuscript in line 105-106.
L83: how were the breeding pairs selected?. Randomly? Expand
Response: Thanks for your suggestion. Pigeons are monogamous, with each pair housed in an individual cage. Offspring are nurtured by their biological parents. Therefore, we randomly assigned 120 pairs of breeding pigeons into 5 treatment groups, with 24 replicates (pairs) per group. We have revised the description, please refer to the manuscript in line 84~86.
L86: ..each breeding pair mated freely?. Not clear. Revise
Response: Thanks for your suggestion. As described in the previous question, pigeons are monogamous, so during the experiment, each pair of breeding pigeons will engage in free mating without any human intervention or artificial insemination.
L88: ..each litter? Revise.
Response: Thanks for your suggestion. Unlike other birds, pigeons lay only 2 eggs per clutch, which hatch after 18 days. They are nursed for 28 days after hatching before the next clutch of eggs is laid.
A MAJOR ISSUE: L150: Statistical analyses – the data were analyzed using one-way ANOVA, and means were compared using the Duncan multiple range test. This is NOT the appropriate analysis. The appropriate statistical method for this design is orthogonal polynomials to examine the linear and quadratic effects of graded levels of riboflavin. As such, differences between individual treatments are NOT important. The authors are using superscripts to separate significance between riboflavin level means, which is not valid. In this statistical model, trends are relevant than treatment differences. Using linear or quadratic effects would have enabled a much focussed presentation. The data should have been be presented accordingly in the Tables and, the 'ABSTRACT", 'RESULTS" and 'DISCUSSION' sections should be re-written. This is a FATAL flaw and must be addressed.
Response: Thank you very much for your suggestions, which will make a crucial contribution to the presentation of our manuscript. We conducted an orthogonal polynomial contrast analysis on all the data to analyze linear and quadratic effects. We supplemented this part of the data in Tables 2 to 7, and included explanations in the abstract, results, and discussion sections. For more details, please refer to the manuscript.
ALL TABLES: individual treatment means +/- SD are not needed. Show the ‘Pooled SEM’ along with L and Q effects. ALSO The numbers and decimals in tables should be follow the rule of: xxxx, xxx, xx.x, x.xx, 0.xxx and 0.0xxx.
Response: Thank you for your suggestions. In the full dataset, we have replaced means +/- SD with Pooled SEM and included linear and quadratic effect analyses. Regarding significant figures, we have made modifications according to the rules. Please refer to the manuscript for details.
TABLE 4 and 5: Absolute weights (g) are irrelevant –these are related to BW. Delete. Give only the % BW data
Response: Thank you for your suggestions. We have re-presented Figure 1 in the form of a table (Table 4), which shows the body weight of squabs at 1d, 7d, 14d, and 28d, with the weight at 28d serving as the basis for the calculation of carcass trait in Table 5. Please refer to the manuscript for details.
FIGURE 1. Present as a Table – why the RB units are different?
Response: Thank you for your suggestions. We have re-presented Figure 1 in the form of a table (Table 4). As described in the materials and methods, Data were tested for normal distribution (Gaussian) using the Shapiro-Wilk normality test. Data with a normal distribution (Shapiro-Wilks test > 0.05) were analyzed using a one-way ANOVA, and means were compared using the Duncan multiple range test (SAS 9.4, 2011, SAS Institute Inc). Adjusted P < 0.05 (Benjamini-Hochberg correction, SAS 9.4) indicated statistical significance; data that did not follow a normal distribution (Shapiro-Wilks test <= 0.05) were analyzed using the Kruskal-Wallis test (SAS 9.4, Z < 0.05 indicated statistical significance).
ALL text need to be revised and discussed as per L and Q responses- NOT by comparing treatment means
Response: We supplemented the linear and quadratic effect analyses in Tables 2 to 7, and included explanations in the abstract, results, and discussion sections. For more details, please refer to the manuscript.

Reviewer 2 Report
Comments and Suggestions for Authors​ 1. What does the research address the main question?
The authors investigated the effects of dietary riboflavin levels on the reproductive performance of pigeons and the growth and carcass quality of offspring to estimate the riboflavin requirements for pigeon breeders.
In the abstract, it is pointed out that the dietary riboflavin levels had no significant effect on body weight, feed intake, egg weight, egg production, and egg fertility (P > 0.05) and then that the pigeons fed a diet without riboflavin had the lowest egg hatchability, egg yolk color, carcass trait, riboflavin status, higher liver weight, and liver index.
Are all the examined parameters important for clients' purchasing decisions, if any, or for pigeon breeders?
2. Do you consider the topic original or relevant in the field?
The topic is original. More experiments are needed on the influence of a diet rich in riboflavin on pigeon egg hatchability, productivity, and carcass quality of offspring compared with chickens and ducks. The conducted research should offer direct optimal riboflavin requirements for pigeon breeders.
The authors point out that breeding practices are geographically different, and we have no explanation for whether they are concerned with applied death or breeding conditions.
3. What does it add to the subject area compared with other published material?
According to the Authors, pigeon diet riboflavin deficiency causes reduced egg hatchability, egg quality, riboflavin status, and quality of eggs and carcasses. It is by other publications for chicken and duck meat.
Proposals for riboflavin concentration in the diet of pigeons are different for hatchability, quality of egg carcass liver yield, and others. It ranges from 4.28 to 13.56. It is a conclusion, but we are still trying to figure out what, in general, the optimal amount of riboflavin in diet pigeons is and if it could work in different geographical regions of China.
4. What specific improvements should the authors consider regarding the methodology? What further controls should be considered?
The methodology is proper and well-presented.
5. Results and discussion
In Fig. 1 and Table 1, the data concerns the effects of riboflavin levels on how many pigeon breeding days.
Meanwhile, the above indicators decreased as dietary riboflavin increased and reached a plateau when dietary riboflavin was above 3.70 mg/kg (can not be seen in Table 1). Any plateau, such expression is good for Figure.
The discussion paragraph comprises the repetition of the study aims.
Riboflavin is essential for poultry production and reproduction. Recommended levels for chickens are 3.6 mg/kg [17] and 8 mg/kg [18]. Ducks require 4 mg/kg [17], 10 mg/kg for meat ducks [19], and 15 mg/kg for duck breeders [19]. It should include information on the recommended level for pigeons.
6. Are the conclusions consistent with the evidence and arguments, and do they address the main question?
The conclusions are consistent with the evidence and arguments but must be more satisfactory. What is the practical information of paper, or what is sufficient riboflavin for the pigeons (not for the culinary part of the carcass or liver…)?
6. Are the references appropriate?
Please check the availability of DOI.
7. Please include any additional comments on the tables and figures.
No comments. They are clear and well-presented. Look for the hint above at point 5.
8. Other comments
Conclusions also should give options for further examinations.
Author Response
- What does the research address the main question?
The authors investigated the effects of dietary riboflavin levels on the reproductive performance of pigeons and the growth and carcass quality of offspring to estimate the riboflavin requirements for pigeon breeders.
In the abstract, it is pointed out that the dietary riboflavin levels had no significant effect on body weight, feed intake, egg weight, egg production, and egg fertility (P > 0.05) and then that the pigeons fed a diet without riboflavin had the lowest egg hatchability, egg yolk color, carcass trait, riboflavin status, higher liver weight, and liver index.
Are all the examined parameters important for clients' purchasing decisions, if any, or for pigeon breeders?
Response: Thank you for your suggestion. The hatchability, growth performance of offspring, and carcass trait are all critical factors for potential buyers. The egg hatchability dictates the number of offspring that can be obtained, and the growth and carcass performance of the offspring directly impact the price of pigeon meat. For instance, in the Chinese market, a 500g squab can be sold for 26 RMB each, while a 400g squab is priced at 22 RMB each. Additionally, squabs with fuller breast muscles fetch higher prices. Currently, there are unified nutritional standards for chickens, ducks, and geese. In recent years, there has been a surge in demand for pigeons, which are now considered the fourth major poultry in China. This growing demand necessitates a standardized production model for pigeon farming. However, there are currently no internationally unified nutritional standards for pigeons. This study explores the effects of dietary riboflavin levels on the breeding performance, offspring production performance, and carcass characteristics of breeder pigeons. It evaluates sensitive indicators for the riboflavin requirements of pigeon breeders through piecewise regression analysis and provides data support for establishing internationally standardized nutritional standards for pigeons. This aims to improve the breeding efficiency and economic benefits of pigeon farming.
- Do you consider the topic original or relevant in the field?
The topic is original. More experiments are needed on the influence of a diet rich in riboflavin on pigeon egg hatchability, productivity, and carcass quality of offspring compared with chickens and ducks. The conducted research should offer direct optimal riboflavin requirements for pigeon breeders.
The authors point out that breeding practices are geographically different, and we have no explanation for whether they are concerned with applied death or breeding conditions.
Response: Thank you for your suggestion. Scientific feeding management is crucial for the pigeon breeding industry, especially from the perspective of diet nutrition. Feed is one of the most fundamental and essential factors in pigeon breeding, directly impacting the birds' growth, reproduction, and meat quality.
Firstly, feed provides the necessary nutrients for pigeons, including proteins, carbohydrates, fats, vitamins, and minerals. Scientifically formulated feed can meet the needs of pigeons' growth, reproduction, and physiological metabolism, contributing to increased hatchability, faster growth of squabs, improved meat quality, and ultimately enhancing breeding efficiency.
Secondly, through scientific feeding management, pigeons can achieve good growth and development. Proper nutritional balance and quantitative feeding can ensure that pigeons receive balanced nutrition, avoiding issues such as slow growth and low body weight, and ensuring that pigeons have good body size and muscle development, thus improving meat quality and quantity.
Furthermore, scientific feeding management can also improve the reproductive performance of pigeons. Providing nutritious and balanced feed can increase pigeons' reproductive capacity, improve hatching rates, and the survival rate of squabs, thereby increasing the number of offspring and the profitability of breeding.
This article conducted relevant experiments from the perspective of riboflavin, showing that the riboflavin level in the feed directly affects the reproductive performance and growth of pigeons. Determining the most appropriate dosage of riboflavin in feed is of great significance for improving breeding efficiency and developing the pigeon breeding industry.
- What does it add to the subject area compared with other published material?
According to the Authors, pigeon diet riboflavin deficiency causes reduced egg hatchability, egg quality, riboflavin status, and quality of eggs and carcasses. It is by other publications for chicken and duck meat.
Proposals for riboflavin concentration in the diet of pigeons are different for hatchability, quality of egg carcass liver yield, and others. It ranges from 4.28 to 13.56. It is a conclusion, but we are still trying to figure out what, in general, the optimal amount of riboflavin in diet pigeons is and if it could work in different geographical regions of China.
Response: Thank you for your suggestion. Currently, there is no established international standard for optimal riboflavin supplementation in pigeons. Based on our experimental results, we aim to provide a reference point for setting future pigeon feeding standards. Since different indicators yield varying estimates, we will present the optimal riboflavin supplementation levels for each indicator. This allows farmers to adjust the supplementation levels according to their farm's specific conditions and the indicators they prioritize. Our experimental design and management model conform to the standard practices in the pigeon breeding industry. The optimal riboflavin requirements identified in this experiment are reliable for other regions.
- What specific improvements should the authors consider regarding the methodology? What further controls should be considered?
The methodology is proper and well-presented.
Response: Thank you very much for your positive feedback on this study.
- Results and discussion
In Fig. 1 and Table 1, the data concerns the effects of riboflavin levels on how many pigeon breeding days.
Meanwhile, the above indicators decreased as dietary riboflavin increased and reached a plateau when dietary riboflavin was above 3.70 mg/kg (can not be seen in Table 1). Any plateau, such expression is good for Figure.
The discussion paragraph comprises the repetition of the study aims.
Riboflavin is essential for poultry production and reproduction. Recommended levels for chickens are 3.6 mg/kg [17] and 8 mg/kg [18]. Ducks require 4 mg/kg [17], 10 mg/kg for meat ducks [19], and 15 mg/kg for duck breeders [19]. It should include information on the recommended level for pigeons.
Response: Thank you for your question. As there is currently no recommended intake information for riboflavin in pigeons, we aim to derive data for riboflavin requirements through our study results. In terms of data presentation, we have incorporated linear and quadratic regression analyses based on the suggestions of Reviewer 1. These analyses will help observe trends in various indicators in response to varying levels of riboflavin in feed. The term "platform phase" refers to a trend where indicators show a delayed response to changes in riboflavin levels, with the minimum riboflavin level corresponding to the highest or lowest range of indicators (inflection point), as shown in Table 8. The linear and piecewise linear models indicate a linear change before the inflection point, followed by stabilization (platform).
- Are the conclusions consistent with the evidence and arguments, and do they address the main question?
The conclusions are consistent with the evidence and arguments but must be more satisfactory. What is the practical information of paper, or what is sufficient riboflavin for the pigeons (not for the culinary part of the carcass or liver…)?
Response: Thank you for your question. The riboflavin level provided in the riboflavin-sufficient diet meets the nutritional needs of pigeons. The specific values are presented in Table 8. According to various indicators, hatchability affects the number of offspring available; growth performance and slaughter performance impact the market value of offspring; yolk color affects egg quality and the transfer of nutrients from parents; liver weight and liver index reflect the health status of squab offspring. These indicators help determine the recommended dosage of riboflavin.
- Are the references appropriate?
Please check the availability of DOI.
Response: Thank you for your suggestion. The journal format does not require the provision of DOIs in the reference section.
- Please include any additional comments on the tables and figures.
No comments. They are clear and well-presented. Look for the hint above at point 5.
Response: Thank you very much for your positive feedback on this study.

Round 2
Reviewer 1 Report
Comments and Suggestions for Authors
Revision is satisfactory
Comments on the Quality of English LanguageMinor editing in syntax is suggested
Author Response
Comments and Suggestions for Authors
Revision is satisfactory
Comments on the Quality of English Language
Minor editing in syntax is suggested
Response: We appreciate the reviewers' recognition of the research content and revisions made in this article. We have corrected some grammatical issues present in the manuscript. Please refer to the manuscript for details.
Reviewer 2 Report
Comments and Suggestions for Authors
The answers of the Authors are not satisfactory:
Are all the examined parameters important for clients' purchasing decisions, if any, or for pigeon breeders?
Currently, there are no internationally unified nutritional standards for pigeons. Therefore, the study examined the effects of dietary riboflavin levels on the breeding performance, offspring production performance, and carcass characteristics of breeder pigeons. Through piecewise regression analysis, indicators for the riboflavin requirements of pigeon breeders were evaluated to establish internationally standardized nutritional standards for pigeons.
The improvement in breeding efficiency has a tremendous economic impact on pigeon farming.
Is it possible to determine the most appropriate dosage of riboflavin in the feed of pigeons?
The authors examine the importance of riboflavin in diet, omitting the answer as precisely the recommended dosage in the nutritional diet for birds.
The Authors present the optimal riboflavin supplementation levels for each indicator. It ranges from 4.28 to 13.56. From a practical point of view, it is hard to apply such information. We should have a close optimum range, for example,8-8.5.
The authors respond that analyses will help observe trends in various indicators in response to varying riboflavin levels in feed. Once more, my question is whether all factors, more or less important, should be considered to give optimal value for riboflavin content in the diet. In many cited publications, you present optimum as one value: recommended levels for chickens are 3.6 mg/kg [17] and 8 mg/kg [18]. Ducks require 4 mg/kg [17], 10 mg/kg for meat ducks [19], and 15 mg/kg for duck breeders [19]. Why is it not possible to give one value to pigeons?
Conclusions also should give options for further examinations.
Author Response
Are all the examined parameters important for clients' purchasing decisions, if any, or for pigeon breeders?
Currently, there are no internationally unified nutritional standards for pigeons. Therefore, the study examined the effects of dietary riboflavin levels on the breeding performance, offspring production performance, and carcass characteristics of breeder pigeons. Through piecewise regression analysis, indicators for the riboflavin requirements of pigeon breeders were evaluated to establish internationally standardized nutritional standards for pigeons.
The improvement in breeding efficiency has a tremendous economic impact on pigeon farming.
Is it possible to determine the most appropriate dosage of riboflavin in the feed of pigeons?
The authors examine the importance of riboflavin in diet, omitting the answer as precisely the recommended dosage in the nutritional diet for birds.
The Authors present the optimal riboflavin supplementation levels for each indicator. It ranges from 4.28 to 13.56. From a practical point of view, it is hard to apply such information. We should have a close optimum range, for example,8-8.5.
The authors respond that analyses will help observe trends in various indicators in response to varying riboflavin levels in feed. Once more, my question is whether all factors, more or less important, should be considered to give optimal value for riboflavin content in the diet. In many cited publications, you present optimum as one value: recommended levels for chickens are 3.6 mg/kg [17] and 8 mg/kg [18]. Ducks require 4 mg/kg [17], 10 mg/kg for meat ducks [19], and 15 mg/kg for duck breeders [19]. Why is it not possible to give one value to pigeons?
Response: Thank you for your suggestion. We have actively considered the reviewer's comments and, based on the characteristics of water-soluble vitamins and the results of this study, we recommend an optimal supplemental dosage of riboflavin in breeding pigeon feed to be 13.6 mg/kg. We have also added the precise recommended dosage of riboflavin in both the abstract and conclusion sections of the manuscript. For details, please refer to the manuscript.
Round 3
Reviewer 2 Report
Comments and Suggestions for Authors
The answers of the Authors are satisfactory:
The Authors, based on the characteristics of water-soluble vitamins and the results of the study, recommend an optimal supplemental dosage of riboflavin in breeding pigeon feed to be 13.6 mg/kg. They have also added the precise recommended dosage of riboflavin in the manuscript's abstract and conclusion sections.
The text is now clearly supported by well-organized tables and figures.
Author Response
Thank you to the reviewers for taking the time to evaluate the content of this manuscript, and for their recognition of our research content and responses.